# High variation in immune responses and parasite phenotypes in naturally acquired *Trypanosoma cruzi* infection in a captive non-human primate breeding colony in Texas, USA

**Angel M. Padilla**[1◉], **Phil Y. Yao**[1◉], **Tre J. Landry**[1], **Gretchen M. Cooley**[1], **Susan M. Mahaney**[2], **Isabela Ribeiro**[3], **John L. VandeBerg**[2], **Rick L. Tarleton**[1]*

**1** Center for Tropical and Emerging Global Diseases and Department of Cellular Biology, University of Georgia, Athens, Georgia, United States of America, **2** Department of Human Genetics, South Texas Diabetes and Obesity Institute, and Center for Vector-Borne Diseases, The University of Texas Rio Grande Valley, Brownsville/Edinburg/Harlingen, Texas, United States of America, **3** Drugs for Neglected Diseases *initiative*, Geneva, Switzerland

◉ These authors contributed equally to this work.
\* tarleton@uga.edu

## Abstract

*Trypanosoma cruzi*, the causative agent of human Chagas disease, is endemic to the southern region of the United States where it routinely infects many host species. The indoor/outdoor housing configuration used in many non-human primate research and breeding facilities in the southern of the USA provides the opportunity for infection by *T. cruzi* and thus provides source material for in-depth investigation of host and parasite dynamics in a natural host species under highly controlled and restricted conditions. For cynomolgus macaques housed at such a facility, we used a combination of serial blood quantitative PCR (qPCR) and hemoculture to confirm infection in >92% of seropositive animals, although each method alone failed to detect infection in >20% of cases. Parasite isolates obtained from 43 of the 64 seropositive macaques were of 2 broad genetic types (discrete typing units, (DTU's) I and IV); both within and between these DTU groupings, isolates displayed a wide variation in growth characteristics and virulence, elicited host immune responses, and susceptibility to drug treatment in a mouse model. Likewise, the macaques displayed a diversity in T cell and antibody response profiles that rarely correlated with parasite DTU type, minimum length of infection, or age of the primate. This study reveals the complexity of infection dynamics, parasite phenotypes, and immune response patterns that can occur in a primate group, despite being housed in a uniform environment at a single location, and the limited time period over which the *T. cruzi* infections were established.

## Author summary

We evaluated naturally occurring infections of *Trypanosoma cruzi*, the causative agent of human Chagas disease, in an indoor/outdoor primate colony at a breeding facility in

**Data Availability Statement:** All relevant data are within the manuscript and its Supporting Information files.

**Funding:** Funded by grant #WT097262 to RLT, JLV and IR from the Wellcome Trust (wellcome.org). The funders had no role in the study design, data collection and analysis, decision to publish, or preparation of the manuscript.

**Competing interests:** The authors have declared that no competing interests exist.

Texas, USA. Using serial quantitative PCR and hemoculture, we confirmed infection in 92% of the 64 seropositive animals, but neither of these two methods confirmed more than 80% of the cases. Parasites by hemoculture fell into two genetic groups (discrete typing units I and IV), and displayed large variation in growth characteristics, elicited cellular and humoral immune responses as well as virulence and drug susceptibility when tested in mice. EKG abnormalities were found in 13 out of 51 qPCR-positive macaques. Our results demonstrate the complexity of these infection parameters in this colony in spite of the uniform and geographically constrained housing conditions of the macaques.

## Introduction

*Trypanosoma cruzi* is the agent of human Chagas disease, the highest impact parasitic disease in the Americas, affecting 10–20 million individuals. Although *T. cruzi* infection has been studied for over a century in its numerous mammalian hosts, including humans, there remain many unanswered questions and misconceptions concerning the factors that determine the control of the infection and the progression and severity of, and the tissues affected by, the disease. Irrespective of the host species, *T. cruzi* is nearly always a persistent infection and disease development is progressive in the absence of effective treatment. Although such treatments exist, few individuals benefit from them for a variety of reasons, including the justifiable concerns of side effects, the documented failure in some cases to cure the infection and the shortage of drugs [1–4]. The misconceptions surrounding *T. cruzi* infection and Chagas disease emanate in part from the largely anecdotal nature of investigations of human infections, the remarkable genetic variability among individual isolates of *T. cruzi*, and the limited number of these genetic variants that have been extensively studied in naturally infected hosts, especially humans.

Natural transmission of *T. cruzi* in humans and domestic animals remains a significant problem in many countries in Latin America and is a growing concern in the southern United States of America [5–7]. *T. cruzi* and insect species that can vector the infection have likely been present in the USA for at least thousands of years [8] and occasional transmission to humans living exclusively in the USA has been documented [9–12]. In contrast to humans, domestic and other animals are quite frequently reported to acquire *T. cruzi* infection, almost certainly via infected insect vectors [13–17]. Included among the animals known to be commonly infected with *T. cruzi* in the USA are non-human primates maintained in breeding and research colonies and in zoological settings. The identification of a large cohort of cynomolgus macaques (*Macaca fascicularis*) with naturally acquired *T. cruzi* infection at the Southwest National Primate Research Center (SNPRC) provided a unique opportunity to study, in a relatively large primate population, the parasitological and immunological parameters associated with the infection and disease.

Herein we report results obtained from a large colony of primate hosts infected with *T. cruzi* via natural environmental exposure and provide the first comprehensive immunological and pathological analysis in conjunction with genotypic and phenotypic characterization of the parasites isolated from these hosts. The study reveals multiple genotypes and a very broad range of phenotypes among parasites isolated from hosts infected in a single, clustered set of housing units. However, the data revealed no association between specific patterns of host immune responses or disease with specific parasite genotypes or phenotypes.

## Results

### Study group

Previous studies documented *T. cruzi* infection among non-human primates housed in outdoor enclosures at the SNPRC where they may have acquired the infection by ingesting infected insects [18,19]. In this study, 64 cynomolgus macaques (*Macaca fascicularis*) of both sexes with positive conventional serology for *T. cruzi* infection were studied. The average age of the animals was 12 years (range 7–22 years) and they were estimated to have been infected a minimum average of approximately 6.4 years (range 1.5–8.5 years; S1 Fig).

### qPCR and hemoculture of serial blood samples failed to detect infection in all seropositive macaques

qPCR was performed on blood samples from the 64 seropositive macaques on up to three different time points over a 3-month period during the spring-summer seasons. This serial qPCR approach yielded a positive result in at least one sample for 51 animals (80%). Out of the 61 animals sampled at least twice, 26 had both samples positive and 22 had only one sample positive. Testing for a third time confirmed the PCR positive result in 7 of 16 animals and detected *T. cruzi* DNA in the blood of 2 of the 13 animals previously testing negative. Cumulatively, for PCR positive animals screened more than once, 83 of those 129 tests (64%) were positive.

For further detection of infection and isolation of parasites from the animals, the red blood cell fraction of whole blood (after the isolation of peripheral blood mononuclear cells (PBMCs) over Ficoll) was cultured and resulted in detectable parasite growth and further cultivation of *T. cruzi* isolates from 43 of the seropositive animals (67%). Qualitative PCR of the hemocultures that lacked visible parasite growth identified 8 additional *T. cruzi* positive hemocultures, bringing the overall detection level for the hemoculture technique to 51 out of 64 samples (80%). Notably, 8 of the 13 consistently qPCR negative, seropositive animals yielded a positive hemoculture with the isolation of viable parasites. The combination of serial qPCR and hemoculture/PCR confirmed *T. cruzi* infection in a total of 59 out of 64 seropositive macaques (92%; Fig 1A). Not surprisingly, blood samples that yielded a positive hemoculture/PCR had on average a higher parasite equivalent per ml value in the qPCR than those that were negative by hemoculture (Fig 1B).

### *T. cruzi*-infected macaques display robust T cell and antibody responses

To further assess the strength and breadth of the antibody responses to *T. cruzi* in these macaques, plasma samples were screened against a panel of 9 recombinant *T. cruzi* proteins as well as a trypomastigote/amastigote lysate using a bead-based multiplex platform [20,21]. All the macaques had high levels of IgG antibodies reactive with the *T. cruzi* lysate which modestly correlated (rs = 0.26; p = 0.04) with parasite loads in blood as determined by qPCR (Fig 2A). As previously demonstrated in humans [20,21] and canines [22] each animal had a distinct profile of antibodies against the antigen panel (Fig 2B). The most frequently detected proteins were the 60S acidic ribosomal subunit protein and a TolT surface protein which were recognized by 77% and 55% of the macaques, respectively (Fig 2C).

All the infected macaques exhibited IFN-gamma-producing cells in response to a parasite lysate and the frequency of responding T cells correlated with their respective antibody responses (Fig 3A) but not with the parasite load (as determined by qPCR; Fig 3B). T cell responses to a pool of macaque CMV peptides were also monitored to assess overall immune competence (as in humans, CMV infection is common in these primates). Although macaque samples displayed a wide range of T cell responsiveness to the CMV peptides, this response did not correlate with the cellular response against *T. cruzi* lysate, suggesting that there was not

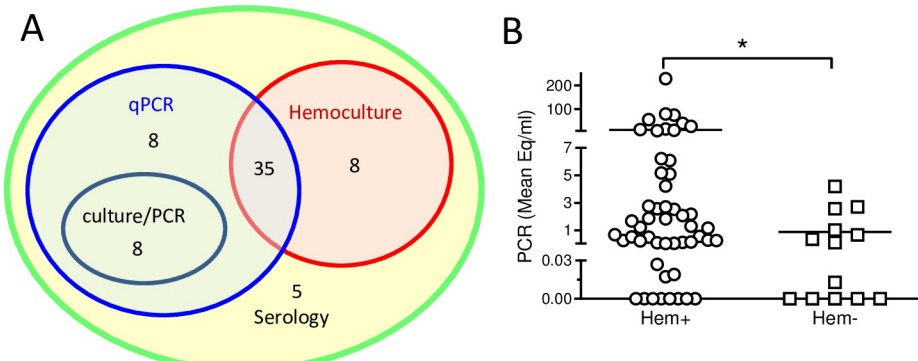

**Fig 1. Serial qPCR and hemoculture confirmed *T. cruzi* infection in 92% of seropositive macaques.** (A) Study group. Peripheral blood samples from 64 macaques positive for *T. cruzi* infection by conventional serology were screened by serial qPCR and hemoculture performed as described in Materials and Methods. Serologically positive macaques were confirmed by blood qPCR (qPCR), visible parasites in hemocultures (Hemoculture) and by qualitative PCR in hemocultures without visible parasites (culture/PCR). (B) Parasite load in blood determined by serial qPCR versus presence or absence of parasite growth in macaque hemocultures. Hemocultures designated positive (Hem+) showed parasite growth or qualitative PCR+ in the culture. Blood samples were analyzed by qPCR at up to three time points, and the mean value of *T. cruzi* equivalents per ml in all the samples for each individual macaque was calculated. Each symbol represents an individual macaque (n = 64). * indicates P<0.05 as determined by Mann Whitney test.

an immunosuppressive effect induced by either *T. cruzi* or CMV that influenced cellular immune responses in general (Fig 3C).

In human *T. cruzi* infection, the intensity and quality of anti-parasite T cell responses decrease with the apparent length of infection in a phenomenon described as T cell exhaustion [23]. However, in this group of infected macaques, a correlation between the strength of the cellular (Fig 4A) or antibody (Fig 4B and 4C) response with the duration of infection was not evident. Similarly, there was not a clear relationship between parasite load in blood by qPCR and the minimum length of the infection (Fig 4D).

Collectively, these immunological analyses document that humoral and cellular immune responses were detected in all study subjects, but that these responses vary greatly in strength and breadth. However, these differences do not appear related to the minimum length of infection and only the antibody response to a *T. cruzi* lysate correlated with the apparent parasite load as assessed by blood qPCR.

## *T. cruzi* infected macaques display low levels of early differentiated (CD45RA⁻CD27⁺CD28⁺) and high levels of late differentiated (CD45RA⁻CD27⁻CD28⁻) memory CD8⁺ T cells

Chronically chagasic patients have been documented to have altered T cell populations consistent with long-term stimulation due to persistent infection [24]. To determine if similar alterations in the overall peripheral T cell populations were present in nonhuman primates with long-term *T. cruzi* infections, we analyzed the different lymphocyte populations in peripheral blood samples of infected animals and compared them with a small group of seronegative macaques from the same colony. Infected animals displayed on average a slightly higher proportion of T effector memory cells (CD28⁻CD95⁺) and a lower proportion of naive T cells (CD28⁺CD95⁻) relative to non-infected controls in both the CD8⁺ and CD4⁺ populations (Fig 5A). Infected animals also displayed a higher proportion of fully differentiated memory (CD45RA⁻CD27⁻CD28⁻) and lower levels of early differentiated memory (CD45RA⁻CD27⁺CD28⁺) CD8⁺ T cells than non-infected macaques (Fig 5B). A higher

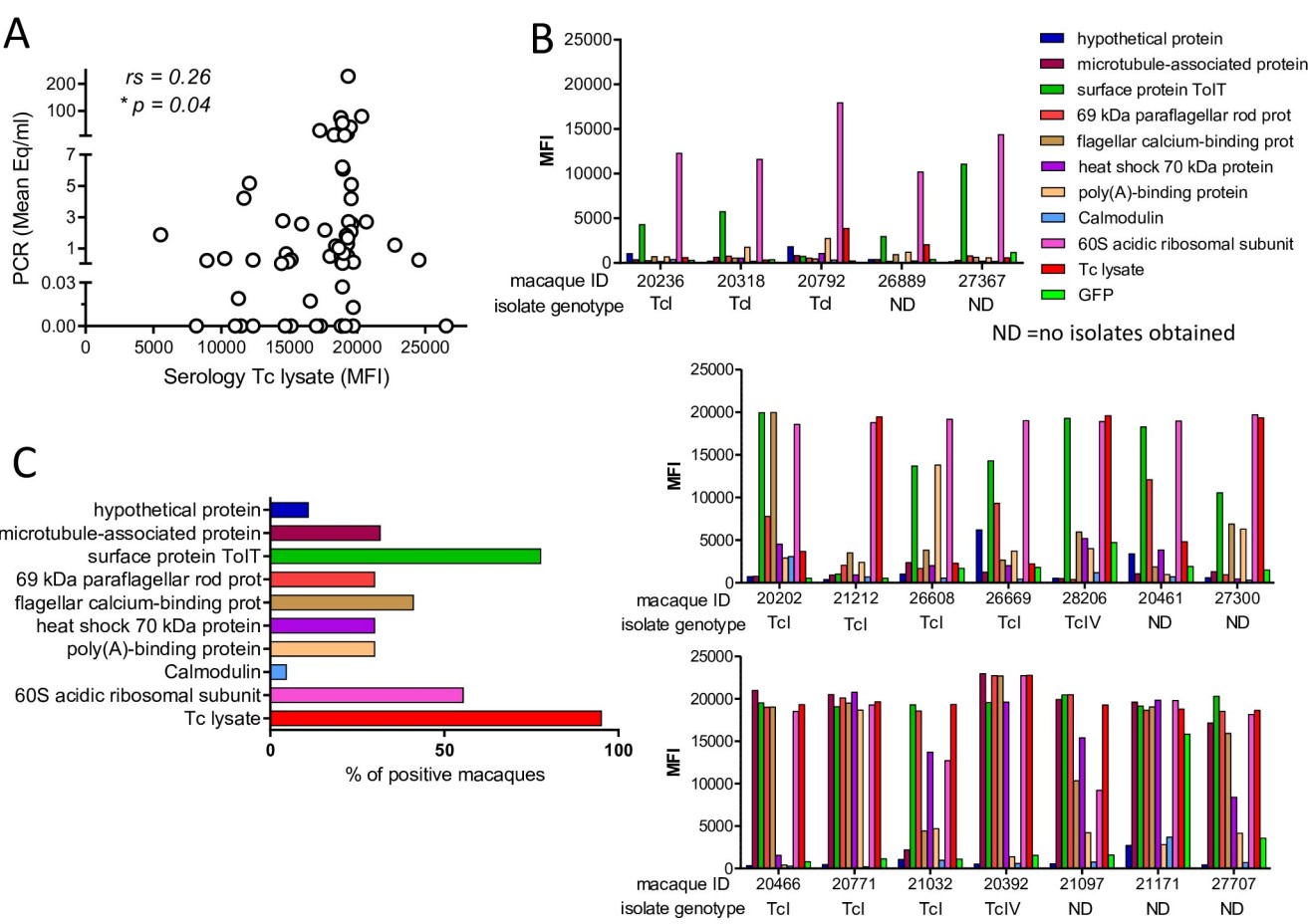

**Fig 2. *T. cruzi* infected macaques display high levels of specific antibodies against parasite antigens.** (A) Reactivity of macaque sera against *T. cruzi* lysate is displayed on the X-axis as median fluorescence intensity (MFI) in a multiplex assay and related on the Y-axis to mean parasite load in blood determined by serial qPCR. Each symbol represents an individual macaque (n = 64). (B) Reactivity against 9 recombinant *T. cruzi* proteins. Each bar represents the median fluorescence intensity (MFI) to the specific antigen. Individual macaques are referred by their identification number on the X axes and the genotype of the parasites isolated from that macaque are noted. *T. cruzi* lysate and green fluorescent protein (GFP) were used as positive and negative controls respectively. Graphs are representative of macaques with weak (top), intermediate (middle), and strong (bottom) reactivity against the recombinant proteins. ND: no parasites were obtained by hemoculture to allow genotyping. (C) Percentage of macaques with a median fluorescence intensity above the negative control (GFP) for each *T. cruzi* recombinant protein. rs: Spearman's correlation coefficient; *p<0.05.

proportion of fully differentiated memory CD4$^+$ T cells was also evident in the infected animals. Thus, the phenotype of the T cell populations reflects *T. cruzi* infection status of the macaques.

## Parasite isolates from infected macaques exhibit significant genetic and phenotypic variation

Genetic characterization of *T. cruzi* isolates from the macaques showed two of the possible 6 genetic lineages previously described in *T. cruzi* [25], TcI and TcIV, with a higher proportion of the former. A genetic pattern corresponding to a mixed infection of both lineages was clearly evident in one animal (Fig 6A). On average, macaques infected with parasites of the TcIV lineage displayed a slightly higher cellular and antibody response against *T. cruzi* lysate (Fig 6B and 6C) although there were no differences in the parasite levels in blood or in the duration of infection between the groups of macaques infected with one or the other lineage (S2 Fig). Parasite lineage also did not track with the diversity or strength of antibody responses (Fig 2B).

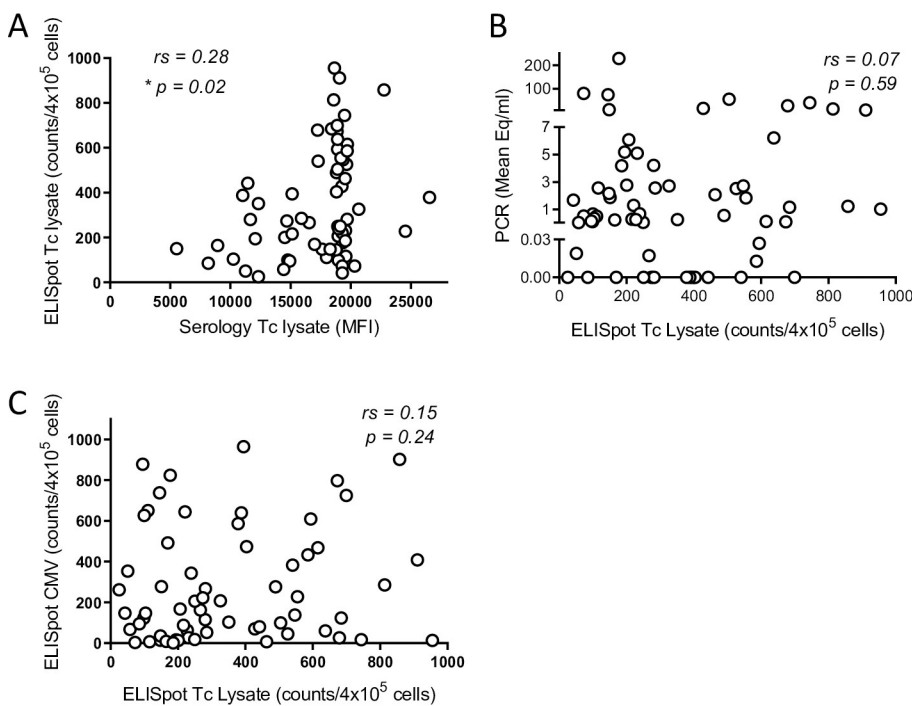

**Fig 3. Infected macaques display a variable T cell response against *T. cruzi* lysate.** (A) Peripheral blood mononuclear cells (PBMC) from seropositive macaques were stimulated overnight with *T. cruzi* lysate and the IFNg response was measured by ELISpot. (A) Cellular and antibody response against *T. cruzi* lysate for individual macaques. (B) Cellular response against *T. cruzi* lysate plotted against the parasite load in blood determined by serial qPCR. (C) The cellular response against *T. cruzi* lysate was compared to the cellular response against cytomegalovirus peptides measured by ELISpot on PBMCs from the same blood sample of individual macaques. rs: Spearman's correlation coefficient; *p<0.05. (n = 64).

To further explore the range of phenotypes in parasites isolated from animals naturally infected in this single housing complex, we infected both C57BL/6J wild-type (WT) and mice genetically deficient in interferon-γ production (IFNγ-KO) with subsets of the isolates. All of these isolates converted from epimastigotes into infective metacyclic trypomastigotes in triatomine artificial urine (TAU) media and all successfully established infections in mice. All WT mice survived acute infection with $10^6$ metacyclic trypomastigotes from 8 isolates, and maintained persistent infections as demonstrated by the detection of *T. cruzi* microsatellite DNA at a range of levels in skeletal muscle, cardiac muscle, and adipose tissues at 120–150 days post-infection (dpi; Fig 7A). Infections in IFN-γ KO mice revealed a wide range of infection intensity among the isolates from macaques (Fig 7B). The majority of the IFN-γ KO mice succumbed to infection by 21 days, however mice infected with some of the isolates survived to ~60 dpi, and IFN-γ KO mice infected with isolate 20392 survived > 90 dpi.

We also measured the ability of isolates to induce peripheral blood $CD8^+$ T cells specific for the immunodominant $CD8^+$ T cell epitope TSKb20, a response we had previously shown to vary depending on the infecting parasite strain [26]. Fig 7C shows that all isolates tested elicited TSKb20-specific T cells although the frequencies observed at 4 weeks post-infection ranged widely.

Lastly, to assess the susceptibility of the isolates to benznidazole (BNZ) treatment, C57BL/6J WT mice were infected, BNZ-treated beginning at ~30 dpi and then immunosuppressed with cyclophosphamide to determine the effectiveness of treatment, as previously described [27]. In 11 of the 16 isolates, BNZ treatment resulted in 100% cure (Fig 7D). However, mice infected

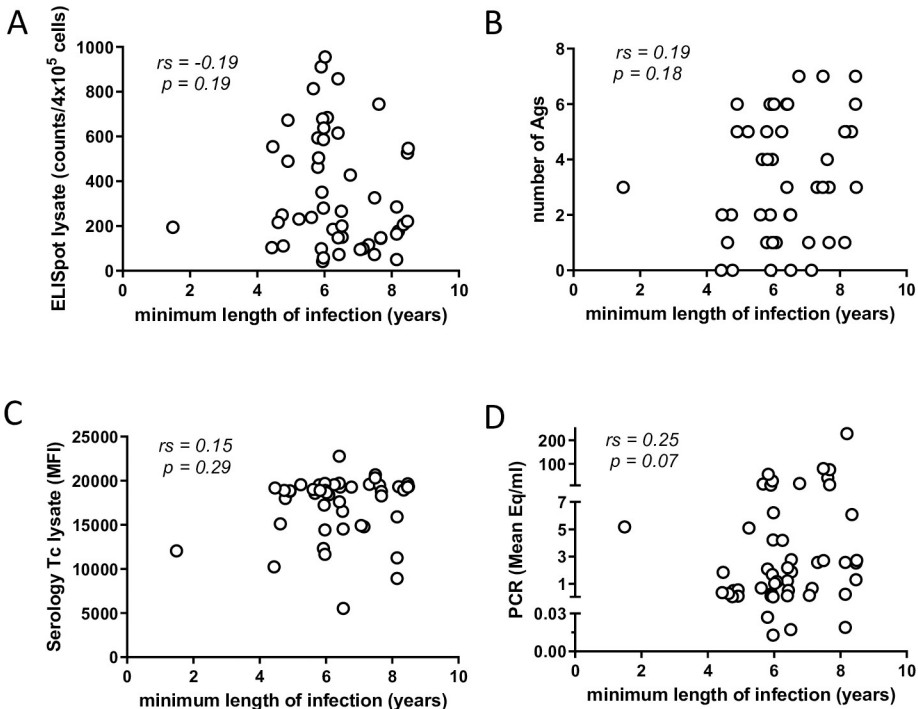

**Fig 4. Minimum length of infection in macaques does not modify immunological and parasitological parameters.**
Cellular (A) and humoral (C) responses against *T. cruzi* lysate for macaques displaying diverse minimum lengths of
infection. (B) Number of recombinant *T. cruzi* antigens recognized by macaques infected for different periods of time
measured by multiplex assay. (D) Parasite load in macaque blood determined by serial qPCR. (n = 50).

with isolates 20202, 20288, 20290, 20349, or 20566 showed a number of treatment failures,
similar to what has been reported with other more "benznidazole-resistant" *T. cruzi* lines [27].

## A proportion of infected macaques displayed EKG abnormalities and a poor health status

The 51 qPCR positive macaques were screened for EKG abnormalities and 26% had at least
one detectable abnormality with 3 animals displaying 2 different abnormalities (Table 1). The
most prevalent cardiac problems were: left anterior fascicular block, sinus bradycardia and
complete right bundle branch block. Notably, these same abnormalities are common in
human chagasic heart disease [28, 29]. There was no association between parasitological or
immunological parameters and the presence of disease (Table 1). Additionally, a small propor-
tion of the animals with a normal EKG profile displayed other health conditions (e.g. heart
murmur or "dull sounds"). When clustered in three different groups: no symptoms, poor
health and cardiac abnormality (Fig 8), the group of macaques with cardiac problems showed
significantly higher antibody responses to *T. cruzi* lysate compared to the group without evi-
dent symptoms, while this trend was not statistically significant for the cellular immune
response. Additionally, no correlations were observed between the parasite load in blood or
minimum length of infection with health status (S3 Fig).

## Discussion

*T. cruzi* infects an extremely wide range of mammalian hosts, in addition to humans, and has
been studied primarily in experimentally infected rodents and occasionally other animals. A

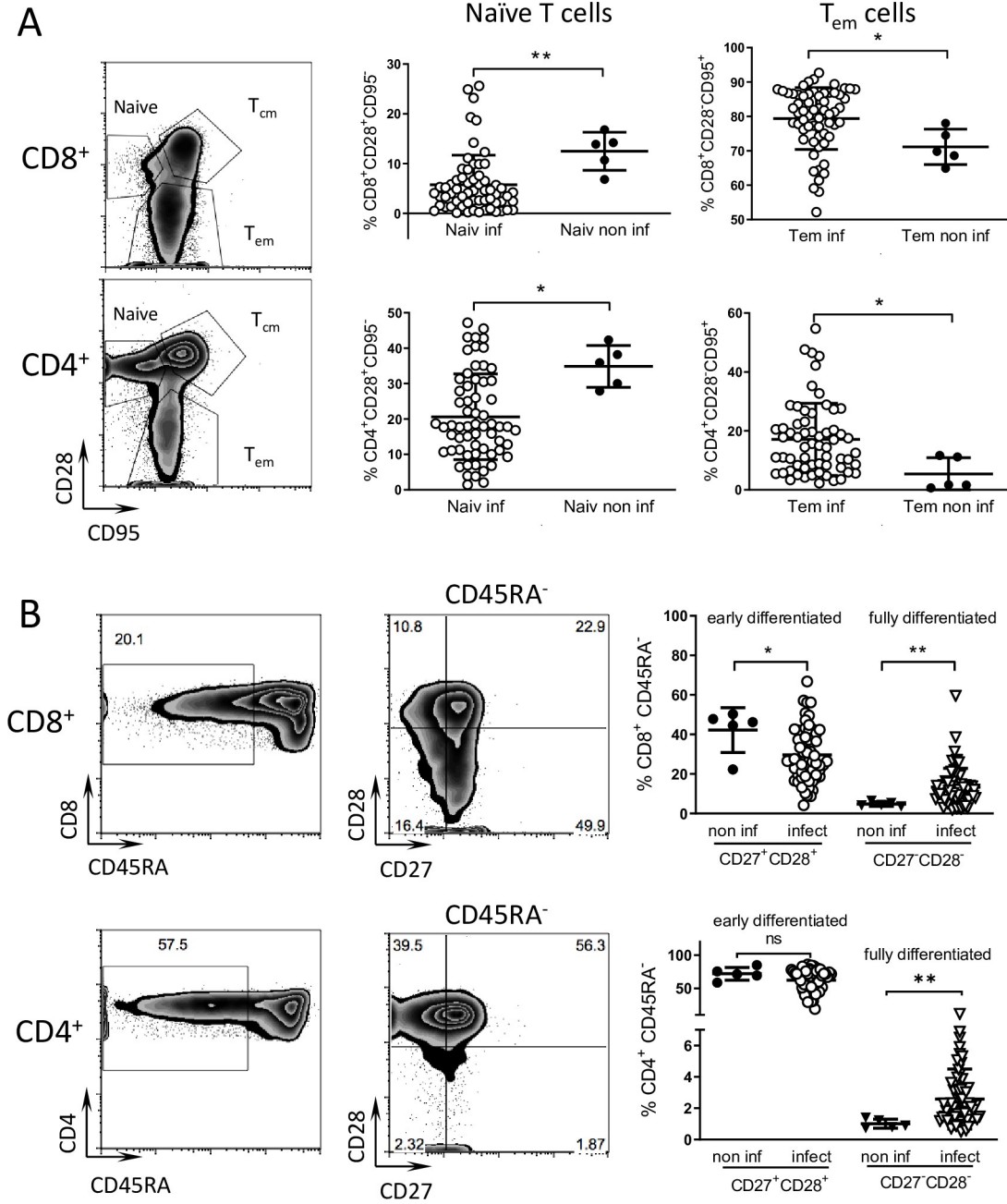

**Fig 5. Phenotype of T cell subpopulation reflects the infected status of seropositive macaques.** Peripheral blood samples were stained for cell surface markers and analyzed by flow cytometry. (A) CD8+ and CD4+ T cell populations were subdivided into naive, T central memory (T_cm) and T effector memory (T_em) subpopulations based on the expression of CD28 and CD95. Naiv inf: naive T cells in infected macaques, Naiv non inf: naive T cells in non-infected macaques, Tem inf: T effector memory cells in infected macaques, Tem non inf: T effector memory cells in non-infected macaques. (B) Early differentiated T cells (CD45RA−CD28+CD27+) and fully differentiated T cells (CD45RA−CD28−CD27−) subpopulations were identified inside CD8+ and CD4+ T cell populations for non-infected (non inf; n = 5) and infected (infect; n = 62) macaques. Representative dot plots of infected macaques are shown. Numbers in the dot plots indicate the percentage of events in the gate or quadrant out of the displayed population. * = p<0.05, ** = p<0.01 by Mann Whitney test.

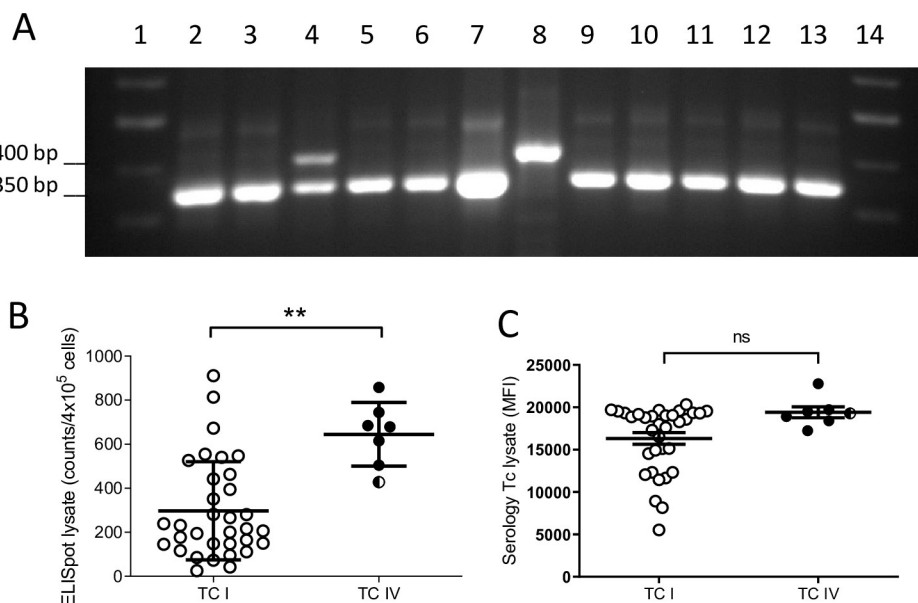

**Fig 6. *T. cruzi* parasites isolated from macaques belong to TcI and TcIV lineages.** Parasite isolates were recovered by hemoculture of red blood cell pellets after PBMC separation by ficoll-hypaque. (A) Representative electrophoresis gel showing macaque isolate genotyping by PCR. TcI and TcIV lineages are identified by a 400 bp and 350 bp band, respectively. Lanes 1 and 14: ladder; lanes 2–6 and 9–13: macaque isolates; lanes 7 and 8: TcI and TcIV reference strains. Note the presence of a macaque with a mixed infection by both genotypes in lane 4. T cell (B) and antibody (C) responses of the macaques infected with parasites belonging to TcIV and TcI lineages. Filled circles: TcIV (n = 6), empty circles: TcI (n = 33), divided circle: isolate with mixed TcI and TcIV genotype (n = 1); ** = p<0.01; ns: nonsignificant by Mann Whitney test.

number of case reports have documented naturally acquired *T. cruzi* in non-human primates and a few additional studies have followed infection of small numbers of experimentally infected non-human primates [30–32]. However, to our knowledge the current study is the first to provide an in-depth study of a primate population naturally and chronically infected with *T. cruzi* in a restricted geographical setting where both a characterization of the parasites involved, and the immunological and clinical status of the hosts, have been examined. Indeed, this level of examination of both parasite and host factors in a defined population of *T. cruzi*-infected hosts has not been conducted in any species, including humans. Thus, this situation of locally acquired, chronic infection in a large number of primates offered a unique research opportunity.

Despite the fact that all animals in this study had acquired *T. cruzi* infection in this highly restricted area, there is considerable parasitological and immunological variation in the infections. At least two genetic lineages of *T. cruzi*, TcI and TcIV, circulate in this setting, however, neither the antibody response, the cardiac health of the macaques, or the parasite load in these animals was associated with the parasite DTU. Furthermore, additional analyses of the parasite lines obtained from these macaques demonstrated that isolates from different animals, but of the same DTU, vary dramatically with respect to induced CD8$^+$ T cell responses and virulence in mice and in relative susceptibility to the anti-*T. cruzi* drug benznidazole. This result contrasts rather strikingly with the suggestions from other studies that the *T. cruzi* DTU determines critical factors about the course and severity of infection, immune response to *T. cruzi*, and disease status [33,34]. It is noteworthy that none of these previous studies compared these parameters among so many isolates collected from such a geographically restricted site as in the current study. These results suggest at a minimum that the parasites of the same DTU and

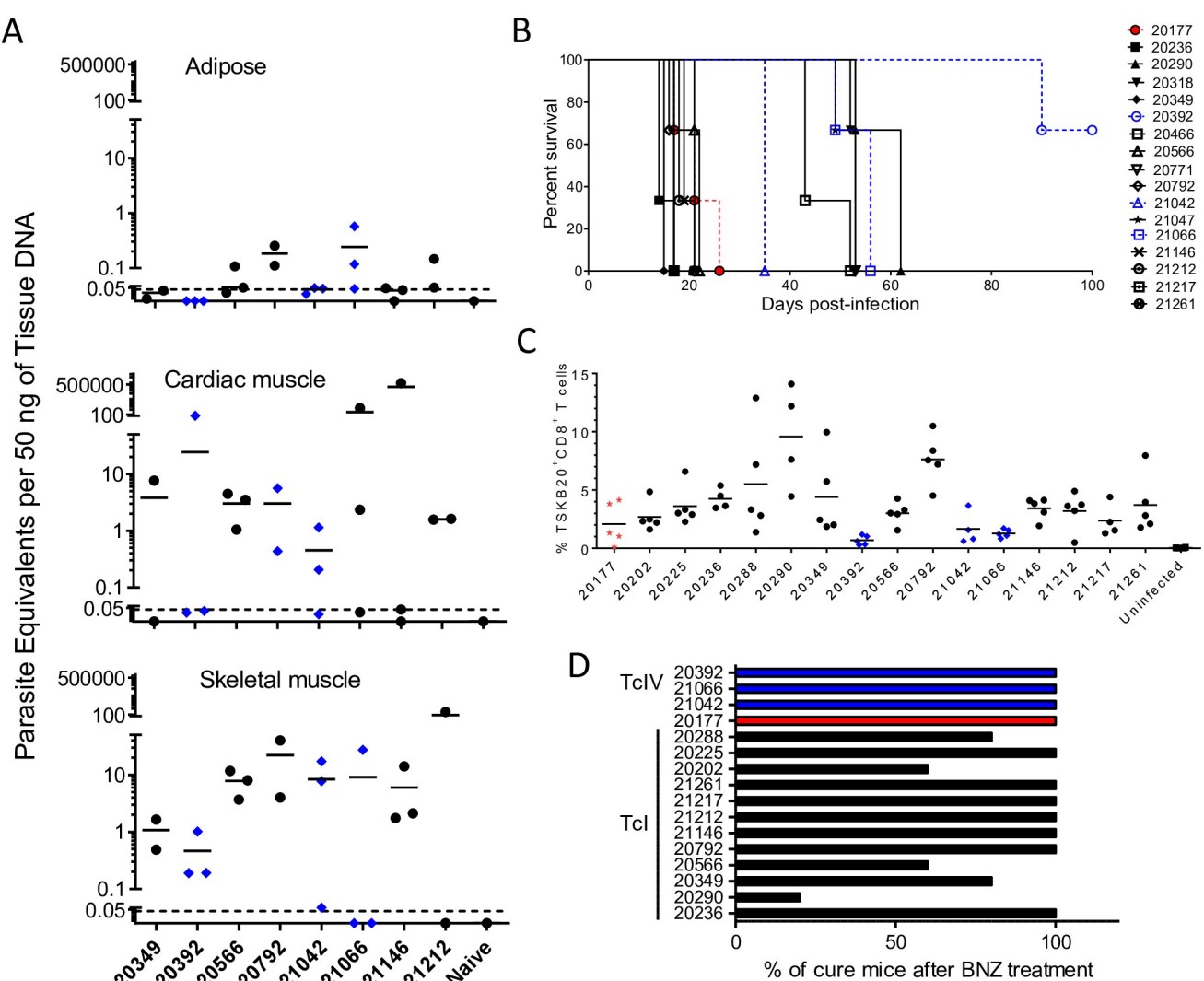

**Fig 7. *T. cruzi* isolates from macaques display a wide phenotypic variation in mice.** (A) *T. cruzi* DNA detection in tissues of C57BL/6 mice infected *per os* (p.o.) with $10^6$ metacyclic trypomastigotes from 8 macaque isolates at 120–150 days post-infection. (B) Survival curve of IFNg-KO mice infected intraperitoneally with $10^6$ metacyclic trypomastigotes from macaque *T. cruzi* isolates (n = 3 mice per isolate). (C) Frequency of CD8+ T cells in blood specific for the immunodominant *T. cruzi*-TSKb20 epitope in C57BL/6 mice orally infected with macaque isolates. (D) Percentage of infected C57BL/6 mice cured after benznidazole treatment, determined by qPCR of skeletal muscle following immunosuppression (n = 4–5 per isolate). DTU: TcI (black), TcIV (blue), or TcI+TcIV (red).

circulating in a narrowly restricted locale have a wide potential with respect to infection outcome. The results of this study indicate that one should expect similar or greater diversity among parasites of the same DTU infecting humans, even in highly constrained geographic settings.

Both of the major genetic lineages of *T. cruzi* previously obtained from hosts in North America were also found in this macaque population and at least one animal had a mixed genotype infection. This latter finding is not unique to this study [35] and indicates either simultaneous transmission of 2 isolates in a single infection episode or more likely, the super-infection of an already infected host (as previously documented in other host species; [36,37]. The finding of a significant population of macaques with naturally acquired *T. cruzi* infection in Texas reconfirms the existence of a highly active transmission cycle in the United States as

**Table 1. Parasitological and immunological parameters in macaques displaying cardiac anomalies.**

|  | Cardiac anomaly[a] | |
|---|---|---|
|  | **positive** | **negative** |
| qPCR (Par Eq/ml) | 7.2 (0.01–55.5) | 13.9 (0.02–228.0) |
| ELISpot (Counts/4x10e5 cells) | 472.1 (42.2–911.0) | 306.0 (50.8–554.8) |
| Serology (MFI) | 19477.1 (17613.5–22772.0) | 16484.9 (78.5–20321.5) |
| Hemoculture[b] | 69.0% (9/13) | 68.0% (26/38) |
| Length of infection (years) | 6.3 (4.7–8.5) | 6.4 (1.5–8.5) |
| Age (years) | 11.7 (7.9–20.4) | 12.0 (8.4–17.4) |

[a] Numbers indicates the average and the range in parenthesis.

[b] For hemocultures the number indicates the percentage of positive and in parenthesis the number of positive cultures out of the total.

well as the diversity and potentially high parasitemia- and disease- generating capacity of parasite isolates circulating in North America. These findings also reinforce the importance of active surveillance for autochthonous cases in humans and particularly in domestic animals in the USA.

In addition to parasite genetics, host genetic factors and the evolving immune responses in infected hosts also contribute to the pattern of infection and disease in *T. cruzi* infection. The health status and infection-related pathology in this group of macaques correlated with the antibody response against a *T. cruzi* lysate (Fig 8). Considering the statistically significant association between the antibody responses and parasite levels in blood across the entire group of macaques (Fig 2), it is tempting to link higher parasite load to poorer health status. However, this direct association was not found (S3 Fig), nor were there statistically significant associations between blood parasite levels, cellular immune parameters and disease presentation. Thus, as expected, the ability of hosts to rigorously restrict parasite levels and also to limit clinical disease is more complicated than can be divined from simply the parasite genotype or from the measurement of numerous, but still quite limited, immune responses at a single point in the infection and in a limited sample of animals with modest disease symptoms overall. These results suggest that caution should also be exercised in drawing broad conclusions about the specific roles of parasite genotypes and general host immune responses with respect to infection/disease outcomes, particularly when based on a relatively small numbers of samples collected over vast geographic areas.

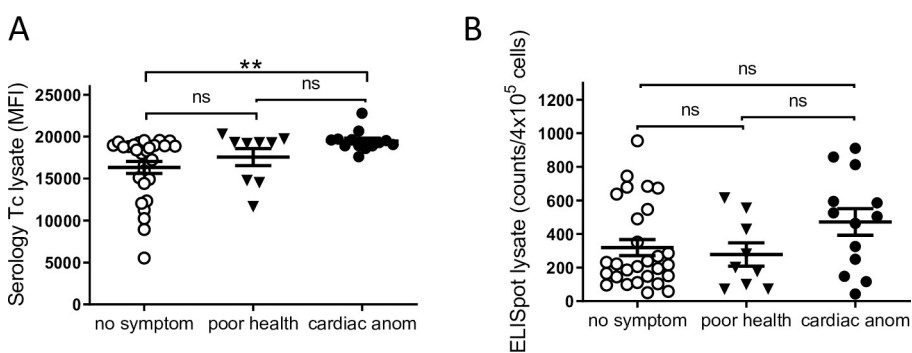

**Fig 8. Macaques with cardiac anomalies show a higher antibody response.** Comparison of antibody (A) and T cell responses (B) against *T. cruzi* lysate in seropositive macaques grouped in: animals displaying no symptoms (no symptom; n = 28), poor general health condition (poor health; n = 9) or cardiac anomalies (cardiac anom; n = 13). ** = p<0.01 by Mann Whitney test.

This study also reemphasizes the limitations of available methods for assessing the *T. cruzi* infection status in chronically exposed hosts. Prior exposure to *T. cruzi* infection can generally be documented by serological assays. However, the scarcity of parasites in chronically infected hosts as well as the immunological memory of that exposure, as demonstrated by the maintenance of sustained high antibody titers, makes determination of active infection–and therefore assessment of spontaneous or drug-induced cure in *T. cruzi* infection–extremely difficult. This situation is the major impediment to the wider use of current therapeutics and the assessment of newly developed ones. PCR is considered to be among the most sensitive of techniques for detection of active infection through the amplification of *T. cruzi* DNA from blood. In this study, the combined analysis by serology, qPCR and hemoculture at 1–3 sampling points over several months again confirmed the inadequacies of parasite detection methods in documenting active infection. Despite analysis of serial bleeds over time, qPCR failed to confirm the infection in some known seropositive animals, including in animals in which parasites were recovered by hemoculture. Thus, despite the powerful amplifying capabilities of qPCR, the low level of parasites in the blood of chronically infected hosts make this technique inadequate for dependable detection of infection or for assessment of infection cure–even when samples are obtained serially over several months. This result strongly suggests that serial qPCR cannot be used as single technique to confirm the infection in scenarios with low parasite levels in blood.

Recovery of parasites from the cultures of red blood cell pellets was surprisingly robust and this protocol could be useful for obtaining parasite isolates from human subjects donating PBMC for immunological studies. Even so, the detection of parasite DNA in hemocultures without parasite growth and the lack of parasite isolation in some qPCR-positive animals shows the limitations of this technique as well. A high positive proportion of hemoculture has also been reported in particular cases of human infection [38]. Although the combination of hemoculture and qPCR confirmed the presence of parasites in the blood of almost all the seropositive macaques studied, the failure of these parasitological techniques emphasizes again the very low level of parasites in the chronic phase of *T. cruzi* infection and the stochastic nature of the techniques to reveal infection, as well as the need to complement infection detection by other methods, including immunological ones. In this regard, nearly all the macaques in the study group displayed a strong antibody response against a parasite lysate and one or more recombinant proteins assays by the multiplex assay, as well as a strong cellular response to *T. cruzi* antigens. Additionally, the proportions of lymphocyte sub-populations reflected the infected status of the animals although with a less strong correlation than has been reported in human patients [24]. Interestingly, infected macaques did not display evidence of T cell exhaustion as described in chronic human patients [39,40] possibly due to the relatively shorter duration of infection of these macaques compared to chronically infected humans previously assessed. Similar findings have been reported in shorter-length human infections (in children) [24].

Collectively, this study highlights the unique and high quality resource that primates with naturally acquired *T. cruzi* infection in the southern of the USA offer for enhanced understanding of Chagas disease, and for further evaluation of techniques for such studies in humans. Perhaps not surprisingly, the results also emphasize the biological complexities of this infection, including host-parasite interactions, and reinforce the known high variation in infection and disease potential of *T. cruzi*, even in a confined geographical space. In addition to the obvious applications of this resource for e.g. testing of new therapeutics, captive non-human primate colonies housed outdoors in some parts of the southern USA provide a unique opportunity to better understand the scope of genetic variation and evolution of *T. cruzi*, and the frequency and consequences of genetic exchange in *T. cruzi* [41–43] under accessible, highly controlled but otherwise natural infection conditions.

## Materials and methods

### Ethics statement

The procedures for maintaining the macaques and blood sampling were approved (#1050MF) by the Institutional Animal Care and Use Committee of the Texas Biomedical Research Institute, the host institution for the SNPRC. All procedures with mice were performed in conformance with protocols and standards approved by the University of Georgia Institutional Animal Care and Use Committee under animal use protocol #A2012 05-003-R2.

### Macaques, blood collection, and EKG

The macaques were housed in mixed-sex breeding groups in indoor/outdoor enclosures constructed of metal, concrete, and cyclone fencing at the SNPRC. The animals were permitted to access both indoor and outdoor portions of the housing. Prior to the initial blood draw, the housing was enclosed with a mesh and the doors were sealed to prevent entry of triatomines, in order to ensure that reinfection did not occur after the initiation of the study. The enclosures contained enrichment devices. Blood was drawn from a saphenous vein after immobilization of the animal by intra-muscular injection of ketamine hydrochloride (10 mg/kg) and valium (5mg), along with inhalation of isoflurane (1.5%). Twelve-lead EKG was performed on the 51 qPCR positive macaques as described by Zabalgoitea et al. [44]. The animal facilities were accredited by the Association for Assessment and Accreditation of Laboratory Animal Care (AAALAC) International at the SNPRC.

### Conventional serology

Serum samples collected from cynomolgus macaques over a period of 7 years were assessed by conventional serology to identify individuals with antibodies against *T. cruzi*. Many of the macaques were sampled and tested at two or more time points, enabling approximate dates of seroconversion from negative to positive to be established. Two assay methods were applied to each sample at each time point. One method was an enzyme-linked immunosorbent assay (ELISA) that detects IgG. Depending on availability at any given time point, the ELISA kits were obtained from one or another of two sources: Bio-Manguinhos (Ministry of Health, Oswaldo Cruz Foundation, Rio de Janeiro, Brazil), and IVD Research, Inc., Carlsbad, CA. The second method in the earlier stages of screening was an indirect immuno-fluorescence assay (IFA) (Bio-Manguinhos) that detects IgG and IgM. That method was replaced by an immuno-chromatographic assay, Chagas Stat-Pak (Chembio Diagnostic Systems, Medford, NY), which detects antibodies directed against several *T. cruzi* recombinant antigens. A sample was designated as positive only if both methods gave positive results. There was a high concordance of results among all four of the assays.

### qPCR for *T. cruzi* in macaque whole blood samples

Whole blood collected in Na$_2$ EDTA tubes was mixed with an equal volume of lysis buffer (6.0M guanidine, 0.2M EDTA, pH = 8.0), and the resultant GE-Blood mixture (GEB) was incubated overnight at room temperature, followed by storage at 4°C. High Pure PCR Template Preparation Kit (Roche Applied Science, Indianapolis, IN) was used to extract DNA from 300 μl of GEB, to which had been added 200 pg of an internal amplification control (IAC, a linearized pZErO plasmid containing a sequence of *Arabidopsis thaliana*; provided by Dr. Alejandro Schijman), as described by Duffy et al. [45,46]. For construction of a standard curve, non-infected monkey blood was spiked with cultured TcI epimastigotes to a concentration of $10^6$ parasite eq/ml, mixed with lysis buffer and IAC, and extracted as described above. Tenfold dilutions of this extract down to a concentration of 1 parasite eq/ml, followed by a

2-fold dilution to a concentration of 0.5 parasite eq/ml, were used to construct the standard curve for the quantification of parasites in each sample. Multiplex TaqMan real-time detection of *T. cruzi* satellite DNA and IAC targets was performed using primer and probe sequences and concentrations, and cycling parameters, in Duffy et al. [46]. Five μl of isolated DNA was analyzed in triplicate using FastStart Universal Probe Master (Roche Applied Science, Indianapolis) in a final reaction volume of 20 μl. Amplifications were carried out in an ABI 7900HT Sequence Detection System (Applied Biosystems, USA). The Tukey method [47] was used to assess the triplicate IAC Ct values obtained from all the samples in each PCR run. If all three replicates of a PCR sample had IAC outliers, as occurred for three of the samples, then the sample was re-extracted and the qPCR procedure was repeated. For all other samples, values with IAC outliers (if any) were discarded, and the mean of the remaining values was recorded as the number of parasite eq/ml. This qPCR method was validated in consultation with Dr. Alejandro Schijman's laboratory (INGEBI-CONICET, Argentina) for precision estimates, anticipated reportable range ($10^5$ to 0.5 parasite eq/mL), limit of quantification (LOQ) (2.46 parasite eq/mL), and limit of detection (LOD) (0.678 parasite eq/mL). For quality control assessment, TcI epimastigotes cultured from one of the macaques were shipped to Dr. Schijman's laboratory, where panels of spiked GEB samples were prepared and returned for blind DNA isolation and qPCR.

## Hemocultures

Following the removal of the ficoll (Sigma-Aldrich, MO) gradient fraction containing the PBMC from 5 ml of whole blood, the remaining red blood cell-containing fraction was culture in liver digested-neutralized tryptose (LDNT) medium at 27˚ C for up to 3 months. Cultures were periodically screened under inverted microscope for motile parasites. Cultures with detectable parasites were propagated in order to obtain and characterize *T. cruzi* isolates. Cultures with no visible parasites were subjected to standard PCR. Briefly, total DNA from hemocultures was extracted using a blood DNA extraction kit (Qiagen, MD) and amplified with OneTaq 2x MasterMix solution (New England Biolabs, MA) mixed with *T. cruzi* specific primers for satellite DNA [48]. PCR products were run on 1.5% agarose gels and imaged in an InGenius gel documentation system (SynGene, MD).

## Parasite isolate genotyping

Epimastigote cultures of parasites isolated by hemoculture from the macaque blood samples were lysed for 30 minutes at 50˚C in lysis buffer (150 mM NaCl, 100 mM EDTA, 100 ug/ml proteinase K, 10 ug/ml RNase A, and 0.5% sodium sarcosinate, pH 8.0). DNA was then extracted twice with phenol/chloroform and ethanol precipitated, then resuspended in 5 mM Tris.Cl, pH 7.5. DNA sample integrity was checked by agarose gel electrophoresis. Discrete typing unit (DTU) determinations were made following the method of Lewis, et al. [49]. Initial discrimination between TcI and TcII—VI was performed by large subunit rRNA (LSU) PCR and to further confirm the presence of TcI and TcIV a second PCR assay was performed to amplify a portion of the mini-exon. TcI isolates amplify a 350 bp product in the mini-exon PCR, whereas TcIV isolates produce a larger, 400 bp, product. PCR products from appropriate DTU of control strains were included in all of the PCR assays to make clearer the sizes of the amplified products in the test samples.

## Cell surface phenotyping

Samples of 100 ul of whole peripheral blood collected in $Na_2EDTA$ were treated with red blood cell lysis buffer and incubated with a mix of fluorescently labeled antibodies against cell

surface proteins. Antibodies used were anti-CD8 PerCP 5.5 (clone RPA-T8), anti-CD4 APC-Cy7 (clone OKT4), anti-CD28 PE-Cy7 (clone CD28.2) from Biolegend, San Diego, CA; anti-CD95 V450 (clone DX2), anti-CD45RA-Allophicocyanin (clone 5H9), anti-CD27-V450 (clone MT271) from BD Pharmingen, BD Biosciences, San Jose, CA; and anti-CCR7-Allophicocyanin (clone 150503) from R&D Systems, Minneapolis, MN. Data from un-fixed samples were acquired in a Beckman Coulter CyAn ADP cytometer and analyzed using the software FlowJo v10.2 (FlowJo, Ashland, OR).

## *T. cruzi* amastigote/trypomastigote lysate

Brazil strain extracellular amastigotes were obtained from trypomastigote cultures induced overnight in pH 5 RPMI 1640 media supplemented with 10% fetal calf serum and 10 mM phosphate citrate buffer. A mix of amastigotes and trypomastigotes was subjected to freeze/thaw cycles followed by sonication. Cellular debris was removed by centrifugation at 12,000 rpm and the supernatant was collected. After filter sterilization, the total protein concentration was determined by Bradford assay.

## Serology testing with recombinant protein multiplex bioassay

Macaque plasma reactivities against 9 recombinant *T. cruzi* proteins and parasite lysate were tested in a multiplex assay previously described for testing human and dog serum samples [20,22]. Briefly, macaque plasma was diluted 1:500 and incubated with a pool of the recombinant *T. cruzi* proteins attached to addressable Liquichip Ni-NTA beads (Qiagen, CA) and *T. cruzi* amastigote/trypomastigote lysate coupled to Carboxy-Ni-NTA beads (Qiagen Inc). Antibody binding was detected with goat anti-human IgG conjugated to phycoerythrin and quantified on a Bio-Plex Suspension Array System (Bio-Rad, Hercules, CA, USA). Weighted median fluorescence intensity (MFI) for samples in duplicate was calculated and the ratio of the specific MFI for each antigen versus a negative control (green fluorescent protein) was estimated for each antigen.

## Interferon-gamma ELISPOT assay

Peripheral blood mononuclear cells (PBMC) isolated by Ficoll gradient were seeded at $4 \times 10^5$ cells/mL and stimulated with 10 mg/mL *T. cruzi* lysate (Brazil strain), cytomegalovirus peptides or 10 ng/mL phorbol-12-myristate-acetate (PMA) and 500 ng/mL Ionomycin for 16 h at 37°C in a 5% CO2 environment; PBMC incubated with complete RPMI media were used as non-stimulated controls. Spot forming cells were automatically enumerated using an ImmunoSpot analyzer (CTL, Cleveland, Ohio). The mean number of spots in triplicate wells was obtained for each condition. The number of specific IFN-γ-secreting T cells was calculated by subtracting the value of the wells containing media alone from the lysate/peptide-stimulated spot count.

## Mice infections with macaque isolates

C57BL/6 and B6.129S7-Ifngtm1Ts/J IFNγ cytokine knockout mice (IFNg-KO) were originally purchased from Charles River Laboratories and The Jackson Laboratory respectively and propagated in the University of Georgia animal facility in microisolator cages under specific pathogen-free conditions. Metacyclic trypomastigotes were differentiated from epimastigote cultures using triatomine artificial urine (TAU) and TAU3AAG and purified by incubating overnight with non-heat inactivated fetal bovine serum as described previously [50,51]. Mice were infected intraperitoneally (i.p.) or per os (p.o.) with $10^6$ metacyclic trypomastigotes and

euthanized between 120–150 days post- infection to collect skeletal muscle, cardiac muscle, and adipose tissues for quantitative Real-Time PCR (RT-PCR) analysis [52].

## Parasite quantification in infected mouse tissues by Quantitative Real-Time PCR

Parasite load was quantified based on the protocol described previously [52]. Briefly, tissues were lysed using tissue lysis buffer consisting of 10 mM Tris–HCl (pH 7.6), 0.1M NaCl, 10 mM EDTA, 0.5% SDS, and 300g of proteinase K/ml. Samples were then heated for 2 h at 55˚C, and extracted with phenol:chloroform:isoamyl alcohol (Fisher Scientific). Cold ethanol was used to wash and precipitate the DNA along with sodium acetate. Each PCR reaction contained 50 ng of genomic DNA, 0.5 μM TcSAT30_F (5'-GGCGGATCGTTTTCGAG), 0.5 μM TcSAT179_R (5'-AAGCGGATAGTTCAGGG), iQ SYBR Green Supermix (Bio-Rad), and nuclease free water (Ambion) to 20 μl. Quantitative PCR was performed using CFX96 Touch Real-Time PCR detection system (Bio-Rad) and analyzed using the Bio-Rad CFX Manager software (Bio-Rad).

## Detection of *T. cruzi* specific CD8$^+$ T cell response in mice

Mouse peripheral blood was obtained by retro-orbital venipuncture, collected in sodium citrate solution, washed, and stained as previously described [53]. Briefly, whole blood was incubated with red blood cell lysis buffer for 5–10 minutes and washed twice with PAB (1x phosphate-buffered saline, 1% bovine serum albumin, and 0.1% sodium azide) solution. Cells were probed with MHC-TSKB20 peptide tetramer complexes conjugated to Brilliant Violet 421 fluorophore and the following labeled antibodies: anti-CD44 PerCP-Cy5.5 (eBioscience, CA), anti-KLRG1 PE-Cyanine7 (eBioscience, CA), anti-CD8a FITC (Accurate Chemical), and anti-CD127 PE (eBioscience, CA). Cells were also stained with anti-CD4 APC-ef780 (eBioscience, CA), which was used as an exclusion channel. A minimum of 500,000 events were collected using a Cyan ADP flow cytometer (Beckman Coulter, FL) and analyzed with FlowJo Version X software (FlowJo, Ashland, OR). MHC-peptide complexes were provided as biotinylated monomers and/or fluorophore-conjugated tetramers by the NIH Tetramer Core Facility at Emory University (Atlanta, GA) and used for identification of CD8$^+$ T cells specific to the TSKB20 (ANYKFTLV/Kb) epitope as described previously [26].

## Sensitivity of macaque isolates to benznidazole treatment

Benznidazole (LAFEPE Medicamentos, Brazil) was prepared by pulverization of one tablet containing 100 mg of the active principle, followed by suspension in distilled water. Mice were drug treated via oral gavage with at 100 mg/kg of body weight once every 5 days for 60 days as described previously [27]. Two to three weeks after the end of treatment, mice were immunosuppressed with cyclophosphamide (200 mg/kg/day) injected i.p. at 2–3 days interval for a total of four doses. Survival and parasitemia were monitored daily in untreated controls beginning 10 days after the start of cyclophosphamide treatment. After parasite detection in blood of untreated controls, all mice were euthanized and tissue samples were collected for parasite load determination by RT-PCR.

## Statistical analysis

Statistical analysis by nonparametric Mann-Whitney test and the nonparametric Spearman correlation coefficient (rs) were performed using the GraphPad Prism 9.0 software. Differences between 2 groups were considered significant at P <0.05.

## Supporting information

**S1 Fig. Macaques in the study had been infected at least for an average duration of 6.5 years.** Male (n = 19) and female (n = 31) macaques in the study had similar ages and displayed comparable minimum length of infection ranging from 1.5 to 8.5 years.
(PDF)

**S2 Fig. Parasite levels in blood and minimum length of infection were similar in macaques infected with either TcI or TcIV.** Parasite load in blood detected by qPCR (A) and minimum length of infection (B) in macaques infected with *T. cruzi* isolates belonging to lineages TcI (n = 33 in A; n = 27 in B) and TcIV (n = 7). ns: non-significant by Mann Whitney test.
(PDF)

**S3 Fig. Health condition of seropositive macaques does not correlate with parasite load or minimum length of infection.** Parasite level in blood determined by qPCR (A) and minimum length of infection (B) in animals displaying no symptoms (no symptom; n = 28), poor general health condition (poor health; n = 9) or cardiac anomalies (cardiac anom; n = 13). ns: non-significant by Mann Whitney test.
(PDF)

**S1 Data. Data set containing all numerical values used to generate Figure panels.**
(XLSX)

## Acknowledgments

The authors thank Jane VandeBerg and associates for establishing the original serological methods and screening the macaque colony; Patricia Frost and associates for managing the macaque colony, collecting the blood samples, and performing the EKGs; Rodolfo Viotti for interpreting the EKG results; Michael Mahaney and Deborah Newman for database development and management; Alejandro Schijman, Carolina Cura, Natalie Juiz, and Juan Carlos Ramirez for guidance and assistance with the blood macaque qPCR; Gloria Bonecini Almeida for providing access to serology kits from Bio-Manguinhos; Todd Minning for the isolate genotyping and Julie Nelson from the CTEGD Cytometry Shared Resource Lab for her assistance in the flow cytometry assays.

## Author Contributions

**Conceptualization:** John L. VandeBerg, Rick L. Tarleton.

**Data curation:** Gretchen M. Cooley, Susan M. Mahaney.

**Formal analysis:** Angel M. Padilla, Phil Y. Yao, Gretchen M. Cooley, Susan M. Mahaney, John L. VandeBerg, Rick L. Tarleton.

**Funding acquisition:** Isabela Ribeiro, John L. VandeBerg, Rick L. Tarleton.

**Investigation:** Angel M. Padilla, Phil Y. Yao, Tre J. Landry, Gretchen M. Cooley, Susan M. Mahaney, John L. VandeBerg, Rick L. Tarleton.

**Methodology:** Phil Y. Yao, Gretchen M. Cooley, Susan M. Mahaney, John L. VandeBerg, Rick L. Tarleton.

**Project administration:** Isabela Ribeiro, John L. VandeBerg, Rick L. Tarleton.

**Resources:** Gretchen M. Cooley, John L. VandeBerg, Rick L. Tarleton.

**Supervision:** Angel M. Padilla, Phil Y. Yao, John L. VandeBerg, Rick L. Tarleton.

**Validation:** Gretchen M. Cooley, Susan M. Mahaney.

**Visualization:** Angel M. Padilla, Gretchen M. Cooley, Rick L. Tarleton.

**Writing – original draft:** Angel M. Padilla, Rick L. Tarleton.

**Writing – review & editing:** Angel M. Padilla, Phil Y. Yao, Isabela Ribeiro, John L. VandeBerg, Rick L. Tarleton.

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
