## [Decision Letter · Decision Letter 0]

15 Feb 2021

Dear Rick

Thank you very much for submitting your manuscript "High variation in immune responses and parasite phenotypes in naturally acquired Trypanosoma cruzi infection in a captive non-human primate breeding colony in Texas, USA" for consideration at PLOS Neglected Tropical Diseases. As with all papers reviewed by the journal, your manuscript was reviewed by members of the editorial board and by three independent reviewers. Overall, the reviewers appreciated the unique opportunity of your group to study the immune responses in a cohort of macaques that had become naturally infected with T. cruzi. The study was considered to be well done and the manuscript clearly written. However, an important critique of the manuscript relates to missing information that would be very helpful for readers to better evaluate the study and its findings. For example, there are references to the health status of the macaques but this evaluation is not backed up by clinical or laboratory findings. Considering that the study aims to correlate parasitological and immunological parameters to clinical findings in this unusual cohort of T. cruzi infected NHP, all relevant information/data should be included as indicated in the reviews.

Based on the reviews, we are likely to accept this manuscript for publication, providing that you modify the manuscript according to the review recommendations. 

Sincerely,

Barbara A Burleigh

Associate Editor

Ana Rodriguez

Deputy Editor

Reviewer's Responses to Questions

**Key Review Criteria Required for Acceptance?**

**Methods**

-Are the objectives of the study clearly articulated with a clear testable hypothesis stated?

-Is the study design appropriate to address the stated objectives?

-Is the population clearly described and appropriate for the hypothesis being tested?

-Is the sample size sufficient to ensure adequate power to address the hypothesis being tested?

-Were correct statistical analysis used to support conclusions?

-Are there concerns about ethical or regulatory requirements being met?

Reviewer #1: - The objectives of the study are clearly stated and addressed. The studied group contained 64 macaques soropositives for T. cruzi infection, housed in an outdoor enclosure at a research facility in Southern US.

- Not all 64 animals were assayed in all techniques through the study. It would facilitate reading if the number of participant animals was indicated in the legends of each graph.

- It should be included the information on the Ig isotypes measured in the study, considering that the isotype gives valuable information about the host's immunity branches involved in their anti-cruzi response.

Reviewer #2: Yes for all questions!

Reviewer #3: The authors performed a series of immune and diagnostic studies that were all performed using high quality and rigorous standards. The authors acquired animal welfare committee approval prior to data collection, and the appropriate veterinary standards were used for data collection. Power calculations were not mentioned in the manuscript, and might have potentially been low due to the relatively small number of infected primates available for investigation. However, the authors did use appropriate statistics and did not make exaggerated claims based on their results.

**Results**

-Does the analysis presented match the analysis plan?

-Are the results clearly and completely presented?

-Are the figures (Tables, Images) of sufficient quality for clarity?

Reviewer #1: - The results are clearly presented, with good quality figures.

- Fig 6A does not show any TcIV results, although in lines 208-209 it is mentioned the authors found TcIV-only samples. The authors should show individual results for all animals in supplementary material.

- The in vivo experiments (Fig 7) did not add much value to the study. The conclusion that "... isolates ... showed a number of treatment failures, similar to what has been reported with other more “benznidazole-resistant” T. cruzi lines [27]" without deeper exploration of the genetics of each isolate andqor the causes for such discrepancies, was not worth the time, resources, and mice invested in these in vitro experiments.

Reviewer #2: Yes for all

Reviewer #3: Results were described and displayed appropriately.

**Conclusions**

-Are the conclusions supported by the data presented?

-Are the limitations of analysis clearly described?

-Do the authors discuss how these data can be helpful to advance our understanding of the topic under study?

-Is public health relevance addressed?

Reviewer #1: - Thirteen macaques were soropositive but had negative qPCR and culture/qPCR results. These macaques were not evaluated by EKG. But then, on line 321, the authors state that "... health status and infection-related pathology ... correlated with the antibody response against a T. cruzi lysate but not with parasite levels in blood or the other immune parameters measured here." Based in their own findings, i.e. that cardiac anomalies correlated only with antibody response, the authors should have included the EKG results from the 13 macaques that were soropositive but had negative qPCR. These 13 macaques is the group that will prove their point from line 321, and these results were not shown. 

- In fact, only 12 out of 64 macaques had EKG exams performed (line 382). I consider this a relevant flaw in the study, since it claims to correlate clinical findings with immunological and parasitological findings (line 289). Furthermore, EKG excerpts from individual macaques must be included as supplementary materials. As it is now, the manuscript requires the reader to accept third party interpretation of the EKG results and the "poor health" status from Fig 8 without having access to clinical and laboratorial parameters used to define "poor health" or the original EKG readings. In this sense, the people involved in performing and interpreting the clinical data and EKGs should share authorship (and accountability) for the findings, instead of just being mentioned in the acknowledgements section (lines 534-535). If this study is reusing previous clinical data acquired routinely at this macaques facility, this must be made clear in the manuscript.

Reviewer #2: Yes

Reviewer #3: The authors' multiple scientific studies resulted in continued support that mammalian host immunologic response to Trypanosoma cruzi infection is complicated and varied. Further, they identified that multiple genetic discrete typing units were present in this focal population, suggesting the presence of co-existence of TCI and TCIV in local USA populations.

**Editorial and Data Presentation Modifications?**

Reviewer #1: Full disclosure of results, supplementary material, and authorships are necessary for supporting the conclusions of the manuscript. Clinical and laboratorial parameters used to define the health status of the macaques should be stated and individual data must be shown, possibly as supplementary material. Individual EKG excerpts for all macaques should be included as supplementary materials.

Reviewer #2: no need to change

Reviewer #3: This article is well written, well supported and ready for publication.

**Summary and General Comments**

Reviewer #1: This study focused on a colony of 64 macaques living in an outdoor enclosure at a research facility in Southern US. These animals have been chronically infected by T. cruzi, what makes them a group of special interest. The study analysed immunological and parasitological parameters in these animals and tried to correlate those findings with their cardiological status. Analysis of serial samples, as it was performed here, are gold standard if we want to fully understand the dynamics of T. cruzi infection. Nevertheless, the study falls short in the characterisation of their clinical status and in the full disclosure of results. This reviewer undertands that revision of results, supplementary material, and authorships are necessary for supporting the conclusions of the manuscript.

Reviewer #2: This an important study that evaluates the immune response induced by differente lineages of T. cruzi. It is unique in the number of samples that were obtained from non-human primates and useful to translate to human disease. The immunological studies are well performed.

Reviewer #3: This investigation took advantage of a convenient sample that is not readily available in nature for inquisition. The authors made several important immunologic and scientific findings, that cumulatively suggest that host immune response to T cruzi infection is quite varied and complicated. One minor recommendation, would be to include the actual months samples were collected and to provide information as to whether the primates were inside versus outside. It was stated that primates were in different conditions, and one could wonder whether those outside during peak triatomine season have the potential for reinfection, which could influence the results. Clearly this would be hard to confirm, but it should be noted in the limitation.

PLOS authors have the option to publish the peer review history of their article (what does this mean?). If published, this will include your full peer review and any attached files.

Reviewer #1: No

Reviewer #2: No

Reviewer #3: No
---

## [Editor Report · Decision Letter 1]

22 Mar 2021

Dear Rick,

We are pleased to inform you that your manuscript 'High variation in immune responses and parasite phenotypes in naturally acquired Trypanosoma cruzi infection in a captive non-human primate breeding colony in Texas, USA' has been provisionally accepted for publication in PLOS Neglected Tropical Diseases.

Best regards,

Barbara A Burleigh

Associate Editor

Ana Rodriguez

Deputy Editor

---

## [Editor Report · Acceptance letter]

29 Mar 2021

Dear Dr. Tarleton,

We are delighted to inform you that your manuscript, "High variation in immune responses and parasite phenotypes in naturally acquired Trypanosoma cruzi infection in a captive non-human primate breeding colony in Texas, USA," has been formally accepted for publication in PLOS Neglected Tropical Diseases.

Best regards,

Shaden Kamhawi

co-Editor-in-Chief

Paul Brindley

co-Editor-in-Chief
